# Self-Compacted Concrete with Self-Protection and Self-Sensing Functionality for Energy Infrastructures

**DOI:** 10.3390/ma13051106

**Published:** 2020-03-02

**Authors:** Alonso Maria Cruz, Puentes Javier

**Affiliations:** Eduardo Torroja Institute for Construction Sciences (IETcc-CSIC), Construction Dpt. Serrano Galvache 4, 28033 Madrid, Spain; javier.puentes@ietcc.csic.es

**Keywords:** SCC, self-diagnosis, electrical resistivity, PZR, CNT, CMF, thermal fatigue

## Abstract

This paper aims to demonstrate the self-protection and self-sensing functionalities of self-compacted concrete (SCC) containing carbon nanotubes (CNT) and carbon microfibers (CMF) in a hybrid system. The ability for self-sensing at room temperature and that of self-protection after thermal fatigue cycles is evaluated. A binder containing a high volume of supplementary mineral additions (30%BFSand20%FA) and different type of aggregates (basalt, limestone, and clinker) are used. The self-diagnosis is assessed measuring electrical resistivity (ER) and piezoresistivity (PZR) in compression mode within the elastic region of the concrete. Thermal fatigue is evaluated with mechanical and crack measurements after heat cycles (290–550 °C). SCC withstands high temperature cycles. The protective effect of the hybrid additive (CNT+CMF) notably diminishes damage by keepinghigher residual strength and lessmicrocracking of the concrete. Significant reductions in ER are detected. The self-diagnosis ability of functionalized SCC isconfirmed with PZR. A content of the hybrid functional additive (CNT+CMF) in the percolation region is recommended to maximize the self-sensing sensitivity. Other parameters as sample geometry, sensor location, power supply, and load level have less influence.

## 1. Introduction

The optimizationof the harvesting and distribution of energy from renewable sources needs specific attention for sustainablegrowth. However, to guarantee the implementation of a determined renewable energy system, which is sufficiently durable for its service life, the parallel advance ofthe construction of the most suitable infrastructure is fundamental. The operating conditions of most renewable energy systems occur in extreme conditions [1]. Offshore wind platforms can be located in different environmental temperatures, from warm to freezing seawater. Windmill towers have to withstand high fatigue loads and extreme environments. Solar power plants and geothermal plants have to endure very high temperatures, and cooling towers are in contact with aggressive acid media. The use of concrete for these types of energy infrastructures is considered to be both challenging and sustainable. Recent experience with these types of concrete constructions hasshown the appearance of early damage that limits their service life and decreases their sustainability. These infrastructurescreate a demand for new concrete technologies which are suitable for the task, such as self-compacted concrete (SCC), together with the incorporation of new additives that improve the performance of the material in aggressive operating conditions and are ableto detectany damage at an early stage.

For solar energy, in the case of solar thermal electricity (STE), the concrete has been considered for the heat storage system, and the severe operating conditions are the high temperatures it has to withstand [2]. The use of concrete for this type of application is a challenge, but its great versatilityworks in its favour when selecting components to improve heat performance, adapting the design to usage needs. The requirements for the concrete vary depending on its role in the system. Until now, it has been used as an isolating material for the foundations of heat storage systems or as a material for thermal storage. The assessment of concrete in the operating conditions of a concentrated solar power plant (CSP) needs to take into account not only the effects of high temperatures, but also the cyclic thermal loads for the heat charge and discharge. While the performance of concrete at high temperatures, such as fire events, is well known and established in standards, there is a lack of knowledge on the response of concreteto long term cyclic thermal fatigue, andonly a few papers have been published [3,4,5,6].The risk of explosions during operation cannot be ignored [7].

Concrete exposed to high temperatures follows a series of physico-chemical changes [8], such as the dehydration of cement paste that results in mechanical losses and the generation of cracks. The selection of adequate concrete components, the cement type (such as Portland cement with the addition of supplementary minerals), and aggregate type (the size distribution and the chemical composition) are fundamental in the design of a concrete that can resist high temperatures [8,9,10].

The major concerns in relation to the use of concrete for heat storage in aCSP is failure due to the risk of explosion, and its stability during the repetitive heatcharge and discharge cycles.

The type of binder and aggregate chosen have been shown to affect the cycling thermal resistance of concrete [5,6] but cracking control and mechanical integrity needs more improvement.

Regarding the design of thermal concrete for STE use, several alternatives have been suggested to improve the resistance of concrete to thermal fatigue: The use of suitable components [4,6], such as cement and aggregates with low expansion and appropriate size distribution and the incorporation of fibers [8,11,12,13,14].

Fiber reinforced concretes (FRC) are increasingly used. The incorporation of fibers to reinforce the concrete matrix is also beneficial for concrete exposed to high temperatures [11]. Polypropylene fibers (PPF) are used to minimize the risk of concrete spalling [12], as theymelt at relatively low temperatures (160ºC), and create new paths for vapor to find a way out, thus relaxing the internal thermal stresses in concrete at high temperatures. In addition to this, the stronger thermal resistance fibers, suchas steel or glass fibers, seem to contribute to keeping the dehydrated cement paste and fiberbound together [8,12,13]. Other typesof fibers which have been used more recently, to reinforce the concrete matrix are carbon fibers (CF) andcarbon nanotubes (CNT) and Carnon nanofibers (CNF), but there is less evidence on their performance at high temperatures [14]. Carbon microfibers (CMF) have shown appropriate thermal stability up to 600 °C, failure is governed by the carbon fiberpull-out from the cement matrix [14]. Besides this, no evidence exists on the performance of FRC with CNT or CMF regarding the deformation of cyclic heat loads. 

Research is needed to demonstrate if advanced concrete technologies, such as self-compacted concrete and the integration of hybrid functional additives such as CNT and CMF, can contribute to the self-protection of the reinforced concrete matrix and improve the performance at the high and cyclic operative temperatures of the STE, to extend the service life of heat storage infrastructures.

As indicated above, several renewable energy systems withstand extreme mechanical operation conditions that result in early damage due to mechanical fatigue. The modification of the electrical resistivity of concrete with the incorporation of conductive additives has added a self-diagnostic property of the mechanical damage of concrete [15,16]. Interest in cementitious materials reinforced with CNT or CF has increased in recent yearsdue to agrowing interest in the capability of concrete structures to self-diagnose.

The self-sensing property in cementitious materials is usually assessed by measuring the electrical resistivity changes in the matrix caused by the incorporation of the conductive additive [17] and with piezoresistivity (PZR) [15,18,19,20,21,22,23,24,25,26,27,28]. PZR is defined as the property that some conductive or semi-conductive materials show when subjected to external strains that induce changes in the material related to the variation of its electrical resistance [18]. PZR is also applicable to cementitious materials that can be enhanced by the incorporation of functional additives to reduce electrical resistivity. The most commonly used self-sensing additives are carbonaceous based ones, such as carbon black (CB) [18], CNT or CNF [19,20,21,22,23], and CF or CMF [24,25,26]. Usually, individual additions of the self-sensing additive are used; the incorporation in mixes such as hybrid additives [27,28] is less common.

Electrical conduction in a cementitious material takes place, first, through the liquid water in the pores, that is, the mobility of ions in the pore solution. This type of conductivity is identified as the ionic conductivity [29] of a cementitious material and is significantly affected by the moisture content in pores [19]. Well-distributedself-sensing additives in the cement paste matrix improves the conductive electrical network of the cementitious material. However, the electrical conduction mechanism of self-sensing cementitious materials is complex and not well understood. The most plausible governing conduction mechanism is identified as beingelectronic (due to the direct contact between the functional additives) and helped by a tunnelling mechanism (conduction occurring between close functional additives which are not in contact) [16,29,30,31]. Nevertheless, the combined contribution of all the conduction systems: Ionic+contacting+tunnelling for electrical conductivity enhancement of cementitious materials with self-sensing additives is the most probable.

Most studies of PZR related to self-sensing assessment have been conducted in cement paste [15,18,19,20,21,22,23] in contrast to those dedicated to mortar or concrete [24,26,28]. Concrete has a more complex and heterogeneous composite structure due to its multi-size components. The coarse and fine aggregates, and theirproportion and size, affect PZR sensitivity. Common aggregates are less conductive than cement paste and behave likes short-circuits in the transfer of the electrical current through the matrix. The cement paste constituted ofmicro-size solid hydrated phases and the pore water solution. The demonstration of self-sensing functionality in concrete is needed to scale-up the self-diagnostic capability to recognizemechanical damage in concrete structures as well as the identification of influencing parameters in the assessment of PZR sensitivity.

Based on the literature from previous analysis, the main objective of this paper is the assessment of functional additives in SCC with carbon nanotubes and carbon microfibers as a hybrid reinforcement of the cement paste matrix from the nano-scale to the micro-scale. The demonstration of the self-sensingcapability and the self-protection performance of the functionalized SCC is made with use ofperformance indicators tailored to each functionality. For self-sensing of concrete electrical resistivity and PZR are usedand forthe enhancement of self-protectionto thermal fatigue, concrete mechanical strength, and cracking are considered.

## 2. Materials and Methods 

SCC have been designed to deal with these objectives. The type of cement used was a CEM II/B-S 52.5, containing mineral additions such as 30% ultra-fine blast furnace slag. This cement was also blended with 20% fly ash (FA). The chemical compositions are included in Table 1. The chemical and physical stability of aggregates is essential for a concrete to perform better in the reduction of thermal fatigue. Therefore, aggregates were selected to improve heat capacity and the specific heat of the concrete [32] while having low thermal expansion. The aggregates, significantly, will influence the final thermo-physical properties of the concrete [32,33] as represented by between 50% and 60% of the total components by weight of the concrete. The maximum aggregate size used was 12 mm because it was consideredthe most suitable size for producing the SCC. Aggregates of a basaltic nature, of 0–6 and 4–12mm, were used. This type of aggregate has a low thermal expansion coefficient, together with adequate thermal conductivity, lowpermeability, and high density. The calcareous aggregates, of 0–6 and 4–12mm, were used due to their low thermal expansion andbecause they havehigher thermal conductivity than basalt. Additionally, the use of a high-densityclinker as an aggregate to cover the size 0–6 was used [34]. For the dosage of the SCC ternary mix aggregates, part of Okamura’s mix design method [35] was used, where the amount of coarse aggregate is set at approximately 50% of the solid volume, in addition to using the amount of fine aggregate at approximately 40% of the mortar volume. The volume of the three types of aggregates was distributed as follows: 60% for calcareous, 30% for basaltic, and 10% for clinker aggregates after a series of tests to allow a balance that guarantees good self-compaction and adequate thermal properties [36]. The main characteristics of the aggregates in accordance withUNE EN 1097-6 are included in Table 2. Limestone filler (LF) was also added to improve the self-compactness of the concrete. Polypropylene fibers (30µmØ and 12mm length) were added to the concretes to control the spalling risk during heating. CMF (Panex 35, 7.2µm Ø and 13mm length) and multiwall CNT (42–65nm Ø and 0.2µm length) for self-functionality enhancement. The main characteristics of PPF, CMF, and CNT are in Table 3. The design was complemented with adjustment of mixtures to fulfill workability, viscosity, and blocking coefficient requirement for self-compactness.

The mixing was carried out in a planetary mixer with a capacity of 120L. The mixing procedurewas: First, all dry components werepremixed for 30 s. PPF and CMF were added into the dry materials. 70% of the mixing water was added until a homogeneous mixture was achieved (approximately 40s). Immediately after, the chemical additives and CNT were added one by one to the evenly distributed remaining mixing water (30%). For the CNT, the pre-dispersion with the water was carried out with magnetic stirring for 15 min. All the components were mixedfor 5 minafter the last component had been added. Table 4 includes the dosages of the concrete components. The self-compactness of the concrete was determined in a fresh state through the J-Ring flow test, UNE EN 12350-12:2011, J-ring SFJ, J-ring t500J, J-ring Final PJ to guarantee a SCC of SF1category.

Cylindrical samples of 75by150mm were prepared for the evaluationof enhancement self-protection to thermal fatigue, while prismatic geometry, samples of 40by40by160 and 70by70by 280mm, were more suitable for sensor location for the self-sensing characterization of the SCC with CNT+CMF.The sample size ratios (lower surface length/maximum aggregate size) were, 6.25, 3.33,and 5.83, respectively which allowedfor adequate component distribution in the SCC matrix.All concrete specimens were cured and kept, until testing, in a moisture chamber, at 20+2°C and 98% R.H.

### 2.1. Thermal Fatigue Test

The thermal fatigue conditions were simulated by designing a heating protocol in three stages, as shown in Figure 1: Stage-1) concrete samples were pre-dried at 105 °C for 72 h to reduce the free pore water. Stage-2) cement paste dehydration, heating from 105 °C to 550 °C, at a slow heating rate of 1 °C/min. Stage-3) concrete thermal fatigue, repetitive heat cycles between 290 °C and 550 °C, as suggested by [6], at a heating rate of 8 °C/min. The total length of each heat cycle was 8 h, inwhich 75 heat charge and discharge cycles (290 to 550 °C) were performed.

The mechanical response of the heated concrete was evaluated in aresidual state, after cooling thesamples to room temperature. The compressive strength and ultrasonic pulse velocity (UPV) were measured for mechanical performance analysis after 0, 1, 5, 25, 50, and 75 heating/coolingcycles for samples of 75by150mm. For crack analysis, slices of 75by25mm were cutand exposed to heat cycles. The crack level of the concrete surface was determined with an optical microscope, Nikon SMZ-2T, coupled to a camera for taking the photos. Two magnifications were used: a) one covering an area of 14.43 mm by 11.25 mm for the definition of the pattern and crack area, and b) another covering an area of 4.81 mm by 3.75 mm for a better measure of the crack width. Using these photos, the evolution of crack width and crack area were analyzed using the photography software GIMP by counting pixels forming cracks.

### 2.2. Self-Sensing Tests

The self-sensing tests were carried out in the functionalized SCC with CNT+CMF. Two methods for showingthe self-sensing sensitivityof the concrete were followed: 1) the measurement of the intrinsic electrical resistivity (ER) was carried out after the concrete had matured for 28 days in the chamber at 98%RH and 20°C, in accordance with the UNE 83988-1standard: Direct method. Figure 2a shows the sample geometry of 75by150mm and the arrangement employed for measurement and 2) thePZR test. The method used was specifically designed toshow the self-sensing capability of functionalized SCC concrete. The PZR protocol was selected after preliminary trials using different arrangements for sensor location (external and embedded) and geometry samples (prismatic and cylindrical). The final PZR method selected used prismatic samples and four embedded sensors (stainless steel meshes) for the electrical field input and output, and data control and collection. Figure 2b shows the geometry with embedded sensors and, on the right, the arrangement and location of the sample for PZR. The distances between the four sensors were 40/80/40 mm (external sensors located on both sides of the two smaller surfaces), 20/40/20 mm, and 35/70/35 mm. Thedepths and distancesbetweenthe sensors are respect the longest face of 160 mm.

The basic data included in the PZR calculation and measured during the test are: Current (A), voltage (V), and load (kN).

The resistivity,ƿ, of the functionalized concrete with CNT+CMF to assess the suitability of PZR is obtained through the Equation (1).
Ƿ concrete (Ohm.m) =R (S/L) = (ΔV/I) (S/L),(1)
with R (Ohm) as the electrical resistance of the concrete, V (mV) as the voltage, and I (mA) as the current. L (mm) as the length between the sensors and S (cm^2^) as the effective area of the sensor.

The basic parameter that reflects the PZR sensitivity of a self-sensing functionalized cementitious material is the fractional change of electrical resistivity (FCR), obtained through the Equation (2)
FCR: (ρ−ρ_0_)/ρ_0_,(2)
with ρ_0_ as the electrical resistivity of the concrete before loading and ρas the electrical resistivity under a specific load.

Compression load changes were used for PZR. A universal compression-testing machine—MEH series—Ibertest MEH—3000—HM 3000kN. For the electrical source, a power supplier HP E3611A DC current, with a voltage range capacity ofup to 35V, connected to a data logger Agilent 349-70A, were used. 

The configuration of the test set-up is shown in the Figure 2c. A voltage gradient was introduced as an electrical field. The current changes (I) were recorded with the external sensors while the potential voltage drop (V) through the two internal sensors. An electrical insulating material was placed on the concrete surfaces in contact with the pressing plates to avoid any interference with the electrical signal and the sensors located on the arrangement of the furthest external side.

When a cementitious material is exposed to an electrical field, the current flows mainly through the pore water solution. The high electrical resistance of a concrete induces a current drop. This phenomenon is identified by some authors as a consequence of the polarization of the ions of the water pore solution under the electrical field [16,18,23]. The drop in current takes a certain amount of time to reach a stateof equilibriumonce the electrical field is applied. To minimize this effect, a stabilization time for the current is recommended before starting with the application of the repetitive compression/decompression load cycles. In addition to this, during the PZR test (loading/unloading) the cementitious material does not always return to its initial resistivity, ρ_0_, as it follows a linear change over time. This phenomenon is also associated with the polarization of the cementitious material. Preliminary tests suggested 30min, at least, for the application of the electrical field before launching the load to achieve a more or less stable equilibrium ofthe current, as shown in Figure 3. A longer stabilization period for the currentwas needed as the polarization voltage applied between thesensors was lower. 

The studies of PZR with cementitious materials [18,20,24] suggest that for a non-destructive PZR method, data should be obtained under loads within the elastic region of the specific cementitious material (<30% compressive stress amplitude of ultimate strength for a reversible elastic deformation of the material compared to the maximum compression or flexural strength [16]). For the selection of suitable compression loads, the characteristic mechanical strength of the concrete was determined: fck: 51Mpa (compression) andE_cs_: 46.04GPa (young dynamic modulus). The maximum resistance reference for testing within the elastic limit (yield point) was set at 0.85fck (43.35 Mpa), from the stress-strain curve. The compression loads used were below the maximum strength in orderto notaffect irreversibly the material and to test the repeatability of the cycles in the same specimen. The loads for the 40by40by160mm were: 10kN (6.25 MPa, 12%Ecs), 15 kN (10MPa, 25%E_cs_), and25kN (15MPa, 33%E_cs_) and for the 70by70by280 75kN (15MPa, 33%E_cs_),per unit area. The compression load mode was selected for the PZR test. The load ratewas0.1kN/s. Table 5 summarizes the parameters of the differentarrangements for the PZR tests.

The PZR performance of the functionalized concretes with CNT+CMF was shown through the analysis of a series of influencing variables, described in Table 5) level of power supplied, 2) maximum load level (% E_cs_), 3) functional additive content, and4) sensor distance and sample size. At least two repetitive PZR cycles were performed to assess each variable in two identical samples; the repeatability of the cycle response for each influencing parameterhas been analyzed.

## 3. Results

### 3.1. Heat Cycles Effect in Mechanical and Crack Nucleation of Concrete with and without CNT+CMF

The variation in the mechanical properties of the SCCsregarding thermal fatigue performance is shown in Figure 4a. Results indicate that the main changes took place during the first heat cycle even with the low heat rate used (1 °C/min). A decrease in compression strength, compared to the initial strengthwas detected. The SCC without the functional additives showed a sharper decay (>50%) while with the functionalized SCC with the CNT+CMF, only 40% was shown. Repetitive heating/cooling cycles (290/550 °C) at a faster heat rate (8 °C/min)didnot affect the residual compressive strength. The UPV measurements, expressed as dynamic young modulus changes, are included in Figure 4b. Decrease of young modulus was detected after the first heat cycle as consequence ofthe dehydration of cement paste and also stabilization with repetitive heat fatigue cycles, which e is in agreement with what is suggested in [6,37]. The stabilization of the mechanical properties of SCC with and without CNT+CMFindicate a suitable thermal fatigue resistance of the SCC concrete designed with low expansion and thermally stable aggregates.

The shrinkage suffered by the cement paste during dehydration, and the changes in the volume of the aggregates during the repetitive charge–discharge heat cycles favour the generation of cracking in the concrete matrix. Figure 5 shows the crack pattern after 75 heat cycles. The crack measurementsindicate that the cracks are micro-sized and are distributed randomly. The cracks run around the interface of the larger aggregates(basalts, in black in Figure 5a), crossing some calcareous (white aggregates, Figure 5c),and through the cement paste. The micro-cracks are more evident in the SCC, initially containing PPF, Figure 5a,b compared to the micro-crack pattern of the functionalized concrete with CNT+CMF, Figure 5c,d.

The evolution of crack growth, identified by the maximum crack width and the percentage of the area of the concreteaffected by the crackafter 1.25, and 75 heat/cool cycles, identified with the digital image system, is presented in Figure 6. The analysis allows the identification of a reduction in the percentage of the surface of the concrete affected by the micro-cracks in the functionalized concrete with CNT + CMF compared to the reference concrete. The maximum crack width is significantly higher in the reference concrete than inthe concrete with CNT+CMF.

### 3.2. Electrical Resistivity Change in Concrete with and without CNT + CMF

The electrical resistance of the functionalized concretes has been considered as a feasible parameter for the self-sensing capability of the SCC. The changes in the electrical conductivity of the functionalized concretes with different hybrid functional additive (CNT+CMF) contents are presented in Figure 7. The results show a clear reduction in electrical resistivity when usingthe combination of 0.2% CNT with CMF. The beneficial synergy of both self-sensing additives, CNT+CMF, as a hybrid reinforcement of the cement paste matrix from the nano to the micro-scale is detected. The minimum electrical resistivity is measured with 3% CMF, up to two orders of magnitude lower than the non-functionalizedSCC (REF).However, the contents of CMF above 1.4%had a detrimental effect on the workability and flowability of the concrete for self-compactness, whichcan affect the distribution of the CMF and CNT in the concrete matrix, and can limit the practical interest in producingfunctionalized SCC technology with self-sensing additives for the further up-scaling of the methodology. The percolation level of the self-sensing SCC was found to have a lower CNT + CMF percentage (0.35% and 0.7%bwb), showinga one order of magnitude decrease in resistivity when compared to the reference concrete.

The heating of the functionalized SCC with CNT+CMF was also affectedby the moisture content in the pores [19]. An increase of two orders of magnitude in resistivity was measured in the functionalized SCC concrete measured at the high temperature of 105°C when the free water was eliminated.

### 3.3. Load-Resistivity Relationship of SCC with CNT+CMF: PZR

As indicated in Table 5, several parameters were used for the evaluation of PZR under repeated compression loads that affecteditsresponse. Figure 8 shows the load/unload cycles and the fractional resistivity change for some variables. Each cycle shows the typical FCR decrease under compression, while the FCR increases during the unload stage, thus confirming the PZR ability of the functionalized SCC.

#### 3.3.1. Influence of the Power Level on PZR Response

The power between sensors was varied, supplying different voltages (2, 5, 10, 15, 20, 30V). The power was applied by decreasing the level from 30volts.The results obtained for the functionalized concrete with 0.2CNT+1.4CMF, load range 15kN (25% E_cs_), and sensor location 40/80/40 mm are shown in Figure 8a. The first aspect which should be highlighted is that a PZR response of the SCC for the different voltages was detected. However, some differences in the range of the FRC of the cycles for the same voltage was detected when low voltages were used, most probably as a consequence of polarization and longer time needed to reach the equilibrium of the current, as demonstrated in Figure 3. The use of voltages higher that 15 V are less noisy FCR changes and more stable responses in cycles, with 20V being selected for later PZR tests.

#### 3.3.2. Effect of the Compression Load Level in PZR

With the aim of detecting whether the change in FCR response also varies in proportion to the load level applied, four-combined loading and unloading cycles from 0 to 10, 15, and 25kN were performed. Figure 8b confirms that the FCR vary proportionally to the load-unload level applied. All load levels show the PZR response of the SCC.

#### 3.3.3. Effect of CMF+CNT Content in PZR

In order to establish the material functionalization level for optimal PZR response, different contents of CNT+CMF around the percolation level (obtained with resistivity, Figure 7) were used. Mixtures with 0.2% CNT and 0.35%, 0.70%, and 1.4% CMF have been tested. The results indicate that even with small variations in CMF, the fraction of the electrical resistivity changes under load. The functionalized concrete with 0.2%CNT+1.4%CMF shows the lowest FCR variation, as this functionalized concrete was after the percolation range. The other CMF contents in the percolation transition range show higher changes in FCR for the same load, as shown in Figure 8c,d. The CMF dispersion is probably less effective, which has a significant effect on the loss of PZR sensitivity, although other factors in the transport of the current cannot be ruled out.

#### 3.3.4. The Influence of Sensor Location

The area and the effective distance of the sensors were considered in the PZR response of the SCC. The distance between the four embeddedsensors was varied as 20-40-20 for the 40 by 40 by 160mm sample, and 35-70-35 for the 70 by 70 by 280mm sample.The aspect ratio as surface/length (S/L) for the current flow was: 40 and 70, respectively.PZR was detected with both sample sizes. The location of the sensors embedded in the concrete guarantees the adhesion of the mesh to the concreteand eliminateany of the effects of de-bonding during testing, and any adverse decay of the PZR response. Thesensor configurationthat has the highest aspect ratiogavelowerFCR changes with a constant load cycle.

## 4. Discussion

### 4.1. The Self-Protection of SCC with CNT+CMF for Thermal Fatigue Stability

When cementitious materials are exposed to temperatures above 100 °C, there is a progressive dehydration of cement paste. This process is accompanied by decreases in mechanical strength, which occurs above 300 °C in most concrete typologies. This phenomenon is well known in fire performance assessment for concrete [8]. The level of strength decrease varies with the composition of the concrete [4,6,7]. In this study, temperatures of up toto 550 °C have been the main element responsible for the dehydration of cement paste and the consequent strength decay detected after the first heat cycle, which is irreversible in further repetitive heat cycles. The stabilization of concrete damage in repetitive heat cycles is of importance if concrete in high temperature environments is of interest, such as its use in concentrated solar power plants for heat storage, where an adequateflow of heat inside the concrete is needed.

The aggregate type and its content and size, have a significant influence on the performance of concrete at high temperatures. Cement paste shrinks due to dehydration while aggregates expandwith heat [8]. These phenomena also drive the generation of cracking. The thermal loads lead to thermal cracking and the thermal cracks are distributed randomly in the bulk of the concrete. The number of thermal cracks is often much higher than cracks in concrete from shrinkage, which can evolve into wider cracks under external tensile loads. The thermal fatigue of concrete at high temperatures, in the SCC designed with special aggregates, is stabilized in the regime of the thermal cycles and attenuation of the concrete damage is detected. The mechanical strength stabilizes in repetitive heat cycles, as shown in Figure 4, which has a similar performance to that found in [3,6]. One relevant aspect in this paper, which has not been analyzed in previous studies, is the cracking response of the concrete inthe thermal cycles. The crack pattern of the SCC stabilizes and keepsits micro-size. In the case of the concrete for a CSP plant, repetitive heat cycles are expected, however, the temperature moves within a high temperature range, 290 to 550 °C. A low detrimental effect for crack generation and propagation isdetected under these extreme heat conditions for the functionalized SCC. Considering the solid components in a concrete, the aggregates represents more than 70% of the total, so they endure most of the volume expansion induced during the first heating to 550 °C. The use of low thermal expansion aggregates can explain the stabilization of crack growth in repetitive heat cycles.

Another relevant aspect which must be highlighted is the more efficient control of cracking with the use of different types and length of fibers to reinforce the SCC matrix, which has shown a positive contribution to the performance of the concrete at high temperatures and in repetitive heat cycles. The bond of the fiber to the cement matrix governs the failure of fiber-reinforced cement composites [8,14]. The reinforcement with PPF disappeared with the temperature increase, as the polypropylene fibers melted above 160 °C, but this process contributed to open the spaces for vapor release, as expected for these typesof fibre which are used for the protection against fire explosion when theconcrete is at high temperatures [11]. The reinforcement using CNT and CMF was beneficial forprotecting the concreteagainst the propagation ofcracks. As a result, the concrete performed better in thermal fatigue cycles. This performance can be attributed to higher stability at high temperatures. The CMF did not decompose at the highest temperature used(550 °C) as observed by SEM in Figure 9a after the first heat cycle (1 °C/min up to 550 °C), which was also detected by [14] with CMF concrete after heating it to 600 °C. A certain bridging of the crack is maintained, after repeated heat cycles, shown in Figure 9b after 25 heat cycles. 

This beneficial effect of the hybrid functional additive CNT+CMF for self-protection at high temperatures and thermal cycles can be seen in Figure 10, showing lower crack widths than the SCC which was not reinforced with CNT and CMF.

The good performance of CMF at high temperatures showsthe suitability of this concrete forenergy infrastructure that operates at extremely high temperatures, asin the case of CSP, if the design of the concrete is appropriately carried out with adequate thermal components as in this study.

### 4.2. The Self-Diagnosis Sensitivity of SCC Reinforced with the Hybrid Additive CNT + CMF

The self-diagnosis property is a function of the test conditions and the optimal dosage of concrete with the self-sensing additives and the electrical resistivity of the concrete matrix, shown in Figure 7 and Figure 8.

In this work, hybrid self-sensing concrete has been used with the aim of covering the deformations in the concrete induced by compressive external loads from the nanoscale to the microscale. Aspreviously mentioned, there is a lack of information on self-sensing capabilities in concrete [24,28]. Concreteis a much more complex system than a simple cement paste asit contains aggregatesof different sizes. Aggregates introduce disturbances in the distribution of currents. The use of concrete generally involves the incorporation of coarse aggregates in the matrix, up to 12 mm in thiswork, and it is a more heterogeneous composite material than cement paste, on which most of the self-sensing studies have been concentrated. The generation of a continuous path for current flow through the self-sensing additives is less favorable in concrete, due to its aggregates, which are less conductive than cement paste. However, the reinforcement of the cement matrix at the microscale with the CMF has contributed to compensate for the negative effect of the aggregates and has favored the flow of current through cement paste, probably with a mixed mechanism for electrical conduction. The water-filled pores are relevant for electrical conduction in the cement paste as tested in [19,22] and the mix of CNT and CMF will favorand contribute to conduction by contacting and tunnelling as predicted in [29,30,31]. In this work, the lack of liquid in the pores at 105 °C has caused a significant increase in electrical resistivity, shown in Figure 7. It is highly likely that this will affect the self-sensing capability of the functionalized SCC detected in saturated conditions. The consequences of temperature increase in PZR needs for a more sophisticated test system that warrantees the maintenance of T during the test. More studies including temperature effect on PZR of concrete are planned in future.

The electrical conduction of conventional concrete is controlled by the water pore content and composition. This type of electrical conduction alone is not able to show the self-sensing performance of concrete in a PZR test, which also occurredin the non-functionalized SCC, the results of which are not included in this work. However, the ionic conduction of an electrical field is probably critical if non-continuous contact conduction can be guaranteed under external loads.

Both an optimal design for the concrete and a suitable PZR test arrangement are neededto demonstrate the self-sensing sensitivity of the concrete and the expectations for the level of the external loads in the operating conditions of a specific structure. Figure 11 shows the influence of several variables related to PZR, thatallow the comparison of the differences in self-sensing sensitivity; while also showing the repeatability of the PZR method used in the reversible range of the mechanical response. A linear region of FCR increase with loading is detected in all the cases studied, in agreement with [27] using cement paste as a self-sensing sensor that explains the FCR which is one order of magnitude higher. However, the length of this linear region is shorter or longer depending on the testing conditions. A change in linearity is detected in some testing conditions in the functionalized SCC, showing a lower increase in FCR with the proportional increase in load. Theeffect of the electrical field conduction and competition between ionic (polarization effect) and contacting and tunnelling electrical conduction probably affects the length of this linear region, between the load and the FCR functionalized concrete response.The presence of aggregates may also effect the situation.

The level of voltage between sensors is relevant both in the sensitivity level and in the repeatability of the FCR response. As seen in Figure 11a, the high voltages (20, 30V) allow higher repeatability from the first cycle as they achievea stable current response more rapidly. Lower voltages need longer to stabilize, and a lack of repeatability in the first cycle of the same voltage was detected. However, after a polarization time, all the voltages showed a similar FCR response. 

Certain scatter in the response of FCR during the load cycle is appreciated, although quite acceptable reversibility between cycles can be observed. The repeatability between the first and second cycle of the FCR signal can also be affected for the rest ofthe variables analyzed in Figure 11. Higher loads and voltages allow better repeatibility. However deeper studies related to repeatibility of the system are of interest, as for instance a large number of FCR load/unload cycles.The first and second cycles practically overlap with the response in FCR, in which a voltage of 20V was used.The load level had a certain influence, shown in Figure 11c, higher loads increased this linear region. With theduplication of the sample size, from 40by40by160 to 70by70by280,the PZR of the functionalized SCC was detected. However, sensor location has a clear effecton the FCR change.In Figure 11b, the increase of loadis expressed as stress for the comparison of the different sample geometries.In the case of the specimens 40by40by160mm and 70by70by280mm with the sensor section/length ratios 40 and 70,the FCR variation is higher when there is a smaller distance between sensors, even though the section of the sensor in the arrangement 35/70/35 was duplicated.

The most efficient parameter in the self-sensing response of the SCC has been shown to be the content of the self-sensing additive, in Figure 11d. Several effects have been highlighted: The highest sensitivity detected for the hybrid additive content in the region of percolationis better than after the percolation level although the lower electrical resistivity of the concrete was measured with the highest CMF content (1.4%). Two reasons could explain this: (i) a lack in the dispersion of the sensing additives and (ii) the higher probability for contact conduction could negatively affect the detection of FCR changes at increasing loads, resulting in lower sensitivity.The need for a certain contact conduction in combination with tunnelling or field conduction, that show lower electrical conductivity in unload conditions, but higher self-sensing under loads. Some authors have also postulated better self-sensing capabilities in the percolation region in cement pastes [30,31].

The fitted functionused for adjustment of the results FCR versus load from Figure 11 was a second order polynomic function. The results are summarized in Table 6. Very good Pearson coefficients have been found in most cases. They are above 0.9, which indicates the soundness of the fit.

The sensitivity of the functionalized concrete to the PZR can be clearly seen in Figure 12. The self-sensing performanceanalysisusedhas been the load sensing coefficient (L_senscoeff_), as described in [19] determined in accordance with Equation (3).
L_senscoeff_= (ρ_0_ − ρ/ρ_0_)/L(3)

With: L as load applied and (ρ_0_-ρ)/ρ the fractional change in resistivity, named FCR.

A linear trend can be seen between L_sencoeff_ and FCR, confirming the highest sensitivity to the hybrid content with the self-sensing additive in the region of percolation, for 0.2% CNT + 0.35% or 0.7% CMF.

## 5. Conclusions

The following conclusions have been derivedfrom the experimental study performed with SCC with CNT+CMF:

The functionalization of SCC has allowedfor the incorporation of new properties to the concrete and the enhancement of is performance in aggressive conditions.

The incorporation of0.2%CNT+1.4%CMF has improved the self-protection of the thermally designed SCC to withstand severe and repetitive thermal fatigue cycles of 290–550 °C, indicated with a mechanical improvement and asignificantly reduced crack pattern.The benefit of the protection of concrete reinforced with CNT+CMF at high temperatures is a significant contribution for using concrete in such environments.

The CNT and CMF have introduced self-sensing sensitivity toSCC.PZR sensitivity has been shown and has contributed to the up-scaling in the detection of mechanical damage of concrete exposed to mechanical stresses. The use of hybrid CNT+CMF allows reductions of up to two orders of magnitudein electrical resistivity in the SCC matrix.The highest sensitivity to PZR is found in the percolation region of the CNT+CMF content.Parameters such as load (in the elastic region), power level and sensor location also affect the PZR response of the self-sensing functionalized SCC, but at a lower scale. The ER significantly increases with the temperature and the PZR response could be affected in the same way, an aspect that needs further study. Moreover, the improvement for better understancing of repeatability of FCR response.

## Figures and Tables

**Figure 1 materials-13-01106-f001:**
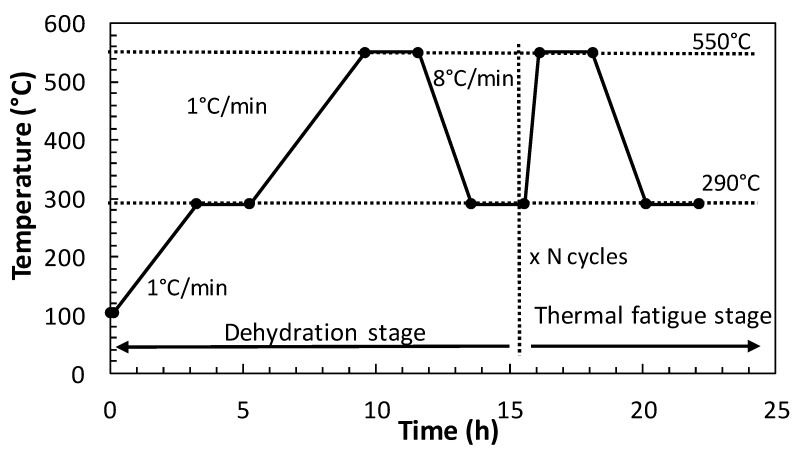
Heat cycle regime for thermal fatigue tests.

**Figure 2 materials-13-01106-f002:**
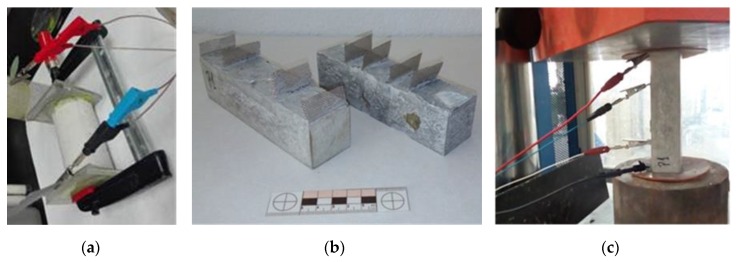
(**a**) electrical resistance, direct method; (**b**)the sample geometry and sensor location; and (**c**) the piezoresistivity (PZR) arrangement.

**Figure 3 materials-13-01106-f003:**
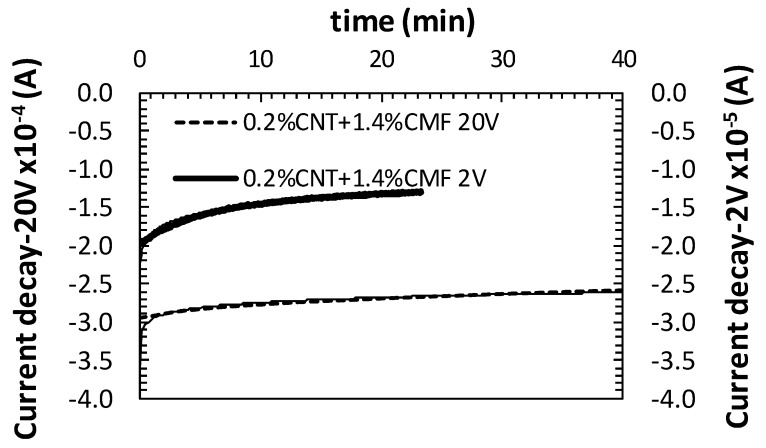
Current decay and stabilization after 2 and 20 V polarization.

**Figure 4 materials-13-01106-f004:**
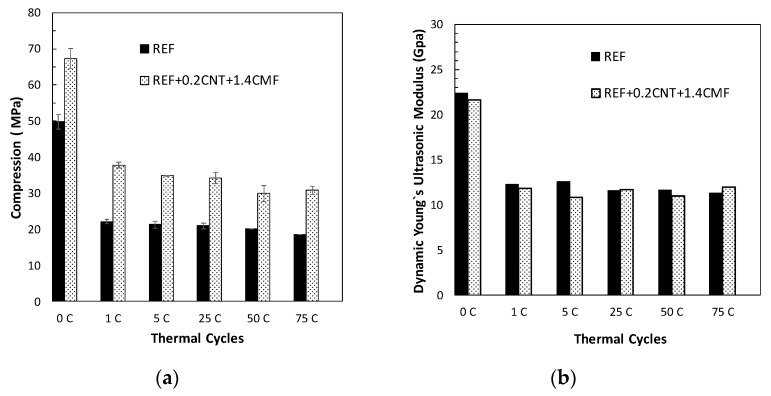
Compressive strength (**a**) and dynamic young modulus (**b**) from SCC with and without CNT+CMF after thermal heat/cool cycles 290 to 550 °C.

**Figure 5 materials-13-01106-f005:**
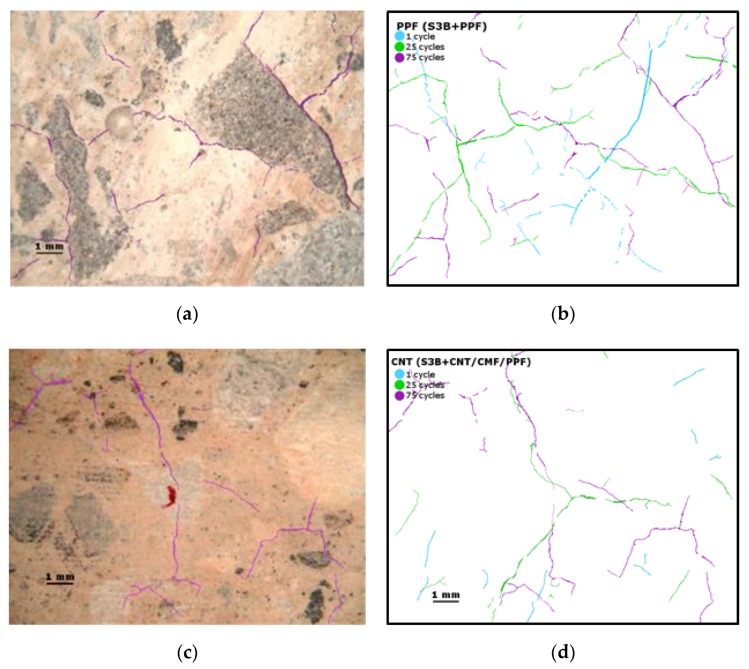
Crack pattern of SCC after 75 heat cycles. (**a**,**b**) without and (**c**,**d**) with 0.2%CNT+1.4%CMF.

**Figure 6 materials-13-01106-f006:**
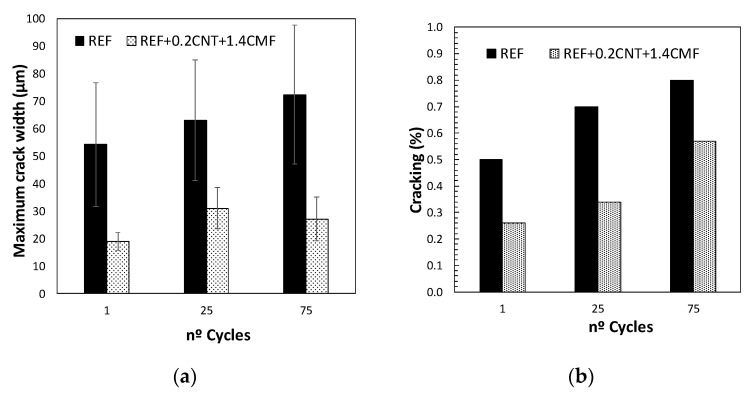
Maximum cracked width (**a**) and the percentage of cracked area after 1.25 and 75 heating/cooling cycles (**b**).

**Figure 7 materials-13-01106-f007:**
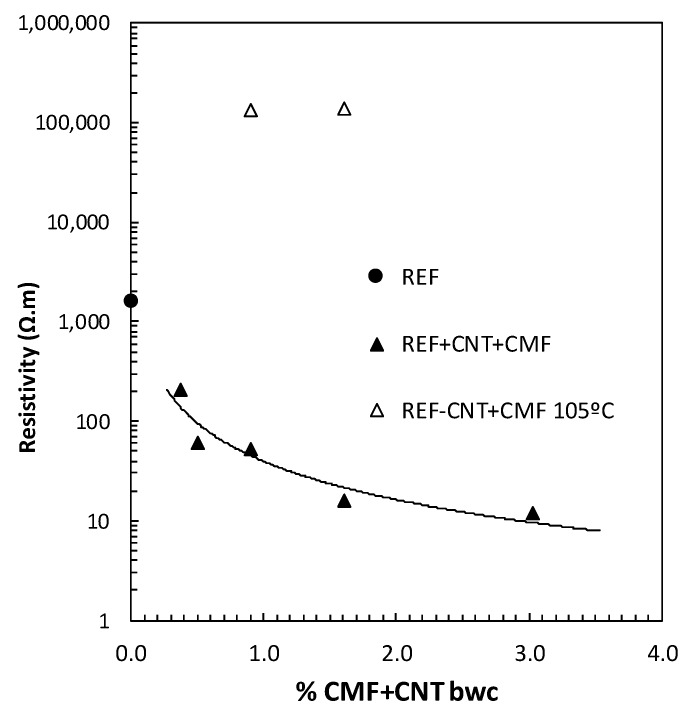
Electrical resistivity changes of SCC with several CNT and CMF contents at 25 and 105 °C.

**Figure 8 materials-13-01106-f008:**
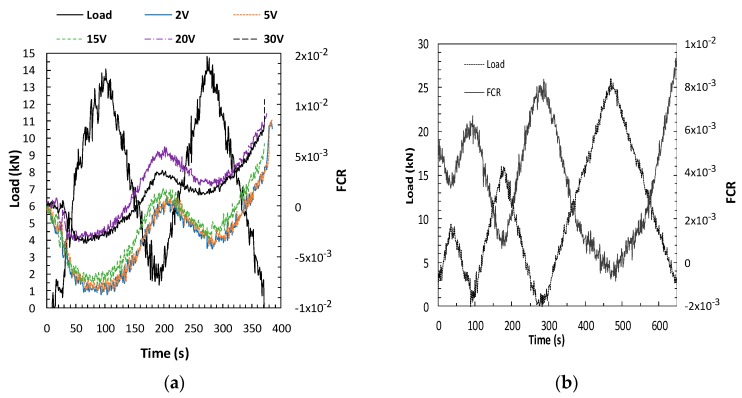
Compression load cycles versusfractional change of electrical resistivity (FCR) for SCC with CNT+CMF. Voltage (**a**), load level (**b**),additive content, 0.2%CNT+0.35%CNT (**c**), and 0.2%CNT+1.4% CMF (**d**).

**Figure 9 materials-13-01106-f009:**
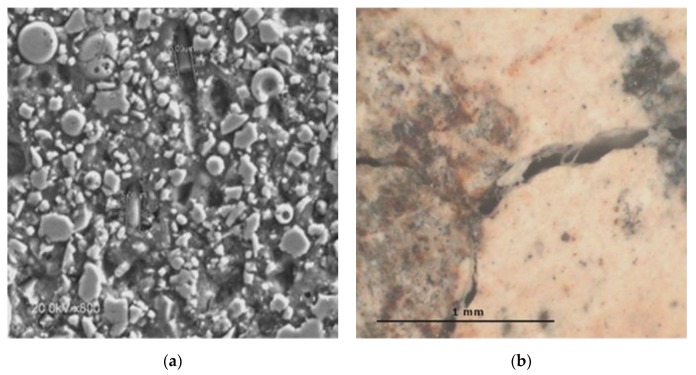
CMF in concrete after heating it to 550°C. (**a**) SEM after the first heat cycle, on the (**b**) after 25 heat cycles.

**Figure 10 materials-13-01106-f010:**
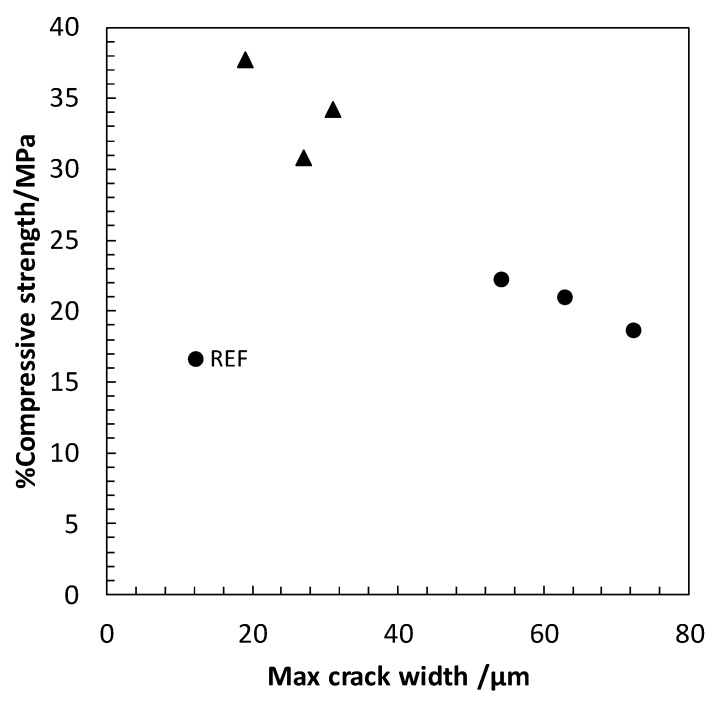
Percentage of residual compressive strength versus crack width for concrete with and without CNT+CMF after 1, 25, and 75 heat cycles (290.550 °C).

**Figure 11 materials-13-01106-f011:**
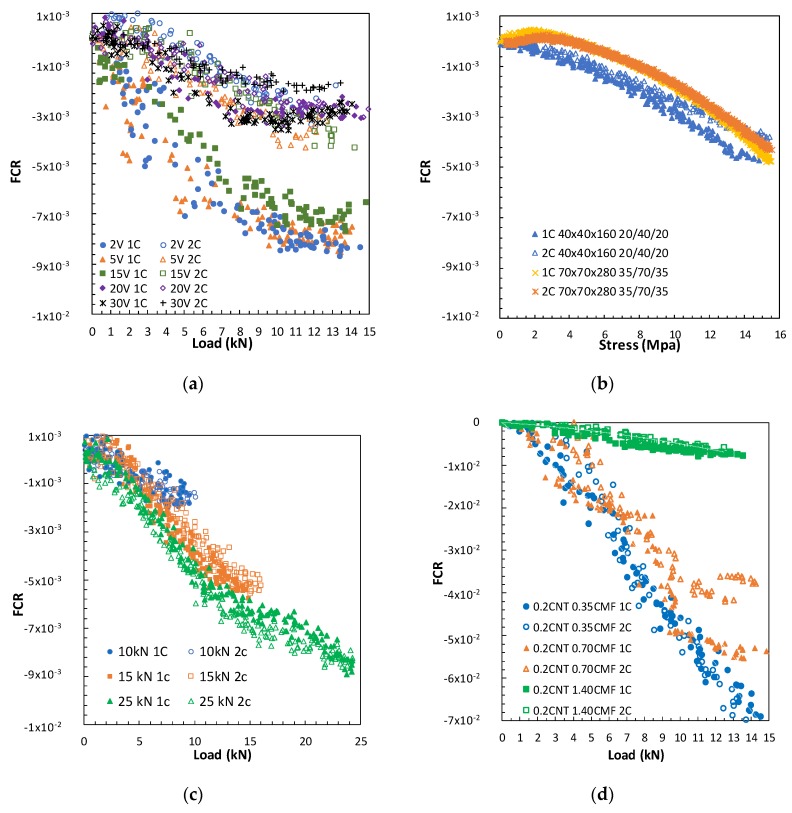
FCR and Load relationship for SCC with CNT+CMF. (**a**) voltage, (**b**) Sensor distance, (**c**) Load level and (**d**) CMF content.

**Figure 12 materials-13-01106-f012:**
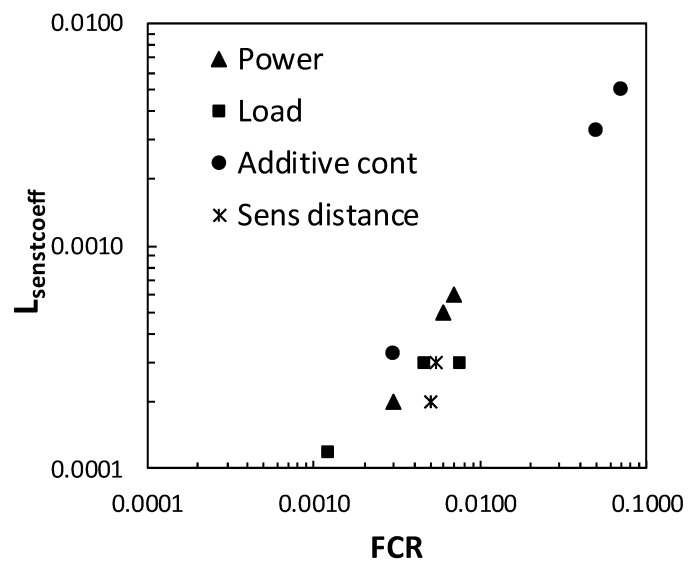
Relationship for Lsencoeff and FCR in the parameters studied for the performance and evaluation of auto-detection.

**Table 1 materials-13-01106-t001:** Chemical composition of the binder.

%	CEM II/B-S 52.5 R	Fly-Ash
SiO_2_	25.7	52.4
CaO	57.8	3.1
Al_2_O_3_	5.9	24.6
Fe_2_O_3_	1.1	8.1
MgO	2.5	1.9
K_2_O	0.5	3.2
SO_3_	2.2	0.38

**Table 2 materials-13-01106-t002:** Aggregates characterization.

Type	Density (mg/m^3^)	Water Absort. Coeff (%)
calcareous	2.69	1.89
Basaltic	2.84	1.55
Clinker	3.1	0.95

**Table 3 materials-13-01106-t003:** Main characteristics of polypropylene fibers (PPF), carbon nanotubes (CNT), and carbon microfibers (CMF).

Type	CMF Panex 35	CNT *	PPF
Length	13mm	0.2–5µm	12mm
Diameter	7.2 µm	49–65nm	31µm
Carbon content	95%	-	
Tensile Strength	3800MPa	-	280–310 MPa
Elastic modulus	242GPa	-	
Resistivity	1.52 × 10^−3^Ω.cm	-	
Density	1.81 g/cm^3^	-	0.91 g/cm^3^
Melting point		-	163–170°C

* CNT were specifically produced in LORCENIS Project by NTUA.

**Table 4 materials-13-01106-t004:** Dosages of self-compacted concretes (SCCs) with low thermal expansion aggregates.

Concrete Components	Thermal Fatigue		Thermal Fatigue & Self-Sensing	
	SCC REF		SCC CNT+CMF	
CEM II/B-S 52.5 (kg/m³)	340		340	
FA (kg/m³)	145		145	
Limestone Filler (kg/m³)	157		330	
CNT (%bwb )	-		0.2	
CMF (%bwb)	-		0.18/0.35/0.70/1.41/3	
PPF (%bwb)	1		0.2	
Calcareous (kg/m³)	500 (0–6mm)	560 (4–12mm)	500 (0–6mm)	531 (4–12mm)
Basalt (kg/m³)	210 (0–6mm)	300 (4–12mm)	300 (0–6mm)	315 (4–12mm)
Clinker (kg/m³)	191 (0–6mm)		191 (0-6mm)	
w/b	0.64		0.6	
Viscocrete 20HE. (%bcm)	1.2		1.2	
Viscoflow 3100 (%bcm)	0.8		0.8	
Sikatell 250 (%bcm)	1		0.9	

**Table 5 materials-13-01106-t005:** Variables for the PZR of self-sensing performance.

Sample Size (mm)	Ad. Content %	Sensor Distance (mm)	Power Supply (V)	Max Load (kN)	Load Rate (kN/s)
40 × 40 × 160	0.2CNT+1.4CMF	40/80/40	2/5/15/20/30	15 (25%E_cs_)	0.1
40 × 40 × 160	0.2CNT+1.4CMF	40/80/40	20	10/15/25 (12,25,22%E_cs_)	0.1
40 × 40 × 160	0.2CNT/ 0.35, 0.7, 1.4CMF	20/40/20	20	15 (25%E_cs_)	0.1
70 × 70 × 280	0.2CNT+1.4CMF	35/70/35	20	25 (33% E_cs_)	0.1

**Table 6 materials-13-01106-t006:** Polynomic fitting of FCR and load relationship, AB constant, and Pearson coefficient variation.

$\mathrm{FCR}=Al^{2}-Bl$	Power Level (V) 2/5/15/20/30	Max Load (kN/%E_cs_) 10/15/25 12/25/33	Additive Content (%bwb) 0.2+/0.3/0.7/1.4	Sensor Distance (mm) 20/40/20 35/70/35
**Constant A**	6/7/4/3/3 × 10^−5^	1.6/1/1 × 10^−5^	7/9/2 × 10^−5^	0.4/0.9 × 10^−5^
**Constant B**	1.4/1.4/1.1/0.6/0.6 × 10^−3^	0.3/2/5.6 × 10^−5^	4.6/4.1/0.3 × 10^−3^	1/0.1 × 10^−4^
**Pearson coeff. (R^2^)**	0.90/0.83/0.96/0.93/0.91	0.74/0.91/0.95	0.97/0.94/0.95	0.98/099

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
