# Peer review of "Self-Compacted Concrete with Self-Protection and Self-Sensing Functionality for Energy Infrastructures"

_materials, 2020, doi:10.3390/ma13051106_

Round 1
Reviewer 1 Report
It is a good paper to discuss the self-compacted concrete, but the authors still need to enhance the introduction part and strengthen your motivation.
Author Response
Reviewer 1
- It is a good paper to discuss the self-compacted concrete, but the authors need to enhance the introduction part and strengthen your motivation
A new version of the introduction has been prepared improving the content and strengthening the motivation of the work
Reviewer 2 Report
The paper “Self-Compacted Concrete with self-sensing functionality for thermal energy infrastructures” proposed for publication by Alonso M.C. and Puentes J. it fits in the field of special concretes such as those self-compacting reinforced with carbon fibers to improve the thermal properties.
The paper is of very good scientific quality, proposes modern methods of analysis and is well documented.
Remarks:
In figure 10 I think that % compressive strength initial / MPa must be passed on the ordinate and max crack width on the abscissa.
In conclusion, in my opinion, the work “Self-Compacted Concrete with self-sensing functionality for thermal energy infrastructures” can be published in Materials.
Author Response
Reviewer 2
- The paper is of very good scientific quality, proposes modern methods of analysis and is very well documented.
In figure 10, I think that % compressive strength initial/MPa mast be passed on the ordinate and max crack width on the abscissa
The figure 10 has been changed according to the reviewer´s suggestion that certainly improve its visualisation of the crack and load relationship
Reviewer 3 Report
Please rewrite and resubmit.
Author Response
Reviewer 3
- Please rewrite and resubmit
Relevant parts of the document have been rewritten and are highlighted in yellow for easier identification. The English has been revised with an English-speaking native, all this has allowed us to better stress the fundamentals and reinforce the advances regarding the state of the art.
Reviewer 4 Report
The evaluated manuscript consists of the description of two groups of tests - thermal fatigue resistance and electrical resistivity. The common element of research is the use of self-compacted concrete reinforced with carbon nanotubes and carbon microfibers. The authors indicate energy infrastructures for heat storage as the place of application of the developed concrete. That place of application is important from the point of view of assessing the scope of performed tests. At the beginning I would like to point out that the performed research is clearly and carefully described in the article and, apart from a few editorial remarks (listed and the end of this review), I have no comments on them.
Thermal fatigue tests are perhaps not very novel, however conclusions and observations are consistent with existing knowledge. Undoubtedly, the authors' original contribution here is common use of carbon nanotubes and carbon microfibers. Especially at high temperatures, the use of carbon fibers seems to be the best choice.
The tests on electrical resistivity I asses much lower. The authors titled them “Self-sensing tests”. This research applies to a small range of compressive stress (up to about 20% strength), insignificant from a practical point of view. Additionally, there is no relationship with the title of the article in which “functionality for thermal energy infrastructures” is indicated. The tests described include only resistivity and load relationship at room temperature. The thermal micro-cracks detected in the first part of the test undoubtedly lead to disturbances in electrical conductivity. Such resistivity changes should be the subject of interest to the authors. In my opinion the article before publication should be supplemented with such research. Their results would be much more interesting in practice. At the same time, the manuscript itself would become more coherent and consistent with the title. In addition, the concept of self-sensing, which in concrete is much more important for tensile stress and cracking, would find sense.
Commenting on the discussion of the results, I would like to point out that the authors do not seem to notice the problem of the repeatability of resistance measurement for cyclic load. For example, in Fig. 8, the FCR increases with each cycle (left-up diagram – second cycle – Load=0; FCR>6.E03). In my experience, this phenomenon, along with moisture sensitivity, is the main problem in testing the electrical conductivity of carbon-fiber reinforced concrete.
Finally some editorial remarks:
Line 113 / Table 2 – density unit g/cm2?
Line 177/Fig.3 – Current decay – which side applies to 2V and which to 20V
Line 195/ Tab. 3 - in my pdf the last column is illegible
Line 235/Fig. 6 title – μcracked (cracked??)
Line 235/Fig. 6 – what is the meaning of the percentage of cracked area - relation of cracks area (their width x length) and the sample area?
Line 261/Fig. 8 – some descriptions are not visible
Line 289 - de-bounding or de-bonding?
Author Response
Reviewer 4
The thermal fatigue tests are perhaps not very novel however; conclusions and observations are consistent with existing knowledge. Undoubtedly, the author´s original contribution here is common use of carbon nanotubes and carbon microfibers. Especially at high temperatures, the use of carbon fibers seems to be the best choice.
- The tests electrical resistivity I asses much lower.
The authors agree that electrical conductivity of a cementitious material vary significantly with the type of system under study. Considering a saturated condition, the resistivity increases from a cement paste to a mortar and concrete, showing this last the higher electrical resistivity, considering the use of the same binder type. The type of cement also influences and binders with high content of SCM increase electrical resistivity. In present paper BFS and FA were used, to a final content of 50%. Besides, in concrete the type of aggregate also influences and the water porosity. In present paper, all samples, results of figure 7, were made in concrete; these aspects would explain the differences with the reviewer experience.
- The authors titled them “Self-sensing tests”. This research apply to small range of compressive stress (up to about 25% strength), insignificant from a practical point of view.
The paper deals with the demonstration of self-diagnosis sensitivity of SCC following two way of analysis, the change in the electrical resistivity and the relationship between the fractional change in electrical resistivity and the external load causing the deformation. Additional interest was to consider the Piezoresistivity (PZR) capability as non-destructive method, what implies the use of loads in the elastic region of the SCC. Different loads levels were evaluated to demonstrate the effect in PZR sensitiveness as reflected in figures 8 and 11.
The authors agree that the sensing property of a concrete that can monitor itself different load methodologies causing deformation can be followed, compression, tension or flexure. Under compression two ways of interest: 1) continuous repeated compression cycles, when compressive stress amplitude is below 30% of ultimate strength a reversible elastic deformation of the material is expected, this is the scope for present paper. 2) Continuous increase of compression until failure or the appearance of cracks. In this last, the PZR as NDT aim is loosed but the detection of un-reversible damage is the interest. Tests until failure were also carried out with this SCC, but are results not considered within the scope of present paper.
- There is not relationship with the title of the article in which “functionality for thermal energy infrastructures” is indicated. The tests described include only resistivity and load relationship at room temperature. The thermal micro-cracks detected in first part of the test undoubtedly lead to disturbances in electrical conductivity. Such resistivity changes should be subject of interest to the authors. In my opinion, the article before publication should be supplemented with such research. Their results would be much more interesting in practice. At the same time, the manuscript itself would become more coherent and consistent with the title. In addition the concept of the self-sensing, which in concrete is much more important for tensile stress and cracking would fine sense.
The title has been change to better adapt with the content of the paper and the motivation. The authors agree with the reviewer on the importance of self-sensing and cracking. The electrical resistivity of a concrete will increase after exposure to high temperatures. Different steps are expected, first reason the loss of liquid water from the pores network and the shrinkage and secondly the dehydration of cement paste and generation of cracking network. The figure 7 has been complemented with electrical resistivity data measured under the high temperature of 105ºC, to reflect the increase of the electrical resistivity due to the loss of liquid water. The consequences of temperature increase in PZR needs for a more sophisticated test system that warrantees the maintenance of T during test. The authors appreciate this comment to address this aspect in future.
- Commenting on the discussion of the results, I would like to point out that the authors do not seem to notice the problem of the repeatability of the resistance measurements for cyclic load. For example, in figure 8 the FCR increases with each cycle (let-up diagram- second cycle- load =0; FCR >6.E03). In my experience, this phenomenon, along with moisture sensitivity, is the main problem in testing the electrical conductivity of carbon-fiber reinforced concrete.
The repeatability of the PZR cycles has been analysed in figure 11 for different variables. The lower repeatability for two cycles has been found when using low voltages in the case of the SCC <15V, as consequence of the longest time needed by the concrete to reach the stability of the current, as demonstrated in Figure 3. However, after the first cycle the response approaches in all voltages. This has been highlighted in the new manuscript in Fig 11. All the PZR test were performed with the saturated state of the SCC to homogenise the moisture content effect in test conditions. The authors agree that the moisture content in pores will influence the PZR response of a self-sensing cementitious material as the electrical resistivity is significantly modified if a decrease takes place.
- Line 113/table 2, density unit g/cm2
Has been corrected: g/cm3
- Line 177/Fig3- Current decay- Which side applied to 2V and which to 20V.
The new figure 3 clarify this aspect
- Line 195/ Tab. 3-in my pdf the last column is illegible.
This has been corrected in the new manuscript
- Line 235/ Fig. 6- µcracked (cracked??).
It has been changed to cracked
- Line 235 Fig. 6- what is the meaning of the percentage of cracked area- relation of cracks area (their width x length) and the sample area?.
This aspect has been better described in the experimental part, in the new manuscript and also clarified in the results description
- Line 261/ Fig. 8- some descriptions are not visible.
The figure caption has been modified and new figures have been prepared of better quality.
- Line 289 – de-bounding or de-bonding?.
It has been corrected to de-bonding
Reviewer 5 Report
1)Line 96, ‘SCC has been designed using thermal aggregates.’. What type of thermal aggregates? In the following description, three types of aggregates were briefly described, but none of them are not thermal aggregates.
2) Basic parameters for every raw material should be clearly presented.
3) Could you please give the mixing proportion design for concretes. How did you consider about the aggregate size distribution?
4) The authors prepared prisms with sizes of 40×40×160 mm for self-sensing performance test. Is it good for sampling and specimen preparation because aggregates have sizes of up to 12 mm?
5) How to determine the flowability of concrete as a SCC?
6) In Fig. 7, more data points should be added to obtain the relationship.
7) The existing or prospective issues should also be given in conclusion parts.
Author Response
Reviewer 5
- The line 96, SCC has been designed using thermal aggregates, what type of thermal aggregates? In the following description, three types of aggregates were briefly described, but none of them are not thermal aggregates.
The authors agree that all the aggregates used show thermal stability in the range of temperatures of the study (up to 550ºC) and low expansion at high temperature expectances to minimise the crack damage due for repetitive heat/cooling cycles. A better description of the type of aggregates and the reason for selection is included. Characteristics of the aggregates are included.
- Basic parameters of every raw material should be clearly presented
The main characteristics of concrete components are included in the new version (tables from 1 to 4).
- Could you please give the mixing proportion design for concrete. How did you consider about the aggregates size distribution
The detail mixing proportion of the concrete components is given in table 4 and the aggregate size distribution method for dosage of the SCC has been indicated in the new version of the paper.
- The authors prepared prims with size of 40x40x160mm for self-sensing performance test. It is good for sampling and specimen preparation because aggregates have sizes up to 12mm?
The authors prepared samples of different sizes and geometries for the PZR study of the SCC with CNT and CMF. Cylinders (75x150mm) and prims(40x40x160 and 70x70x280mm are considered in present paper. The prism geometry is more suitable if embedded sensors are used, as in present paper. The lower dimension of the sample (40mm) is more than double to the max aggregate size (12mm). The ratio minimum surface size/max aggregate size=3.33. The high fluidity of the SCC has also contribute for a well distribution of the aggregates in the samples.
- How to determine the flowability of concrete as a SCC?
Indications of the standard followed and parameters analysed are indicated in the new manuscript
- In Fig 7., more data points should be added to obtain the relationship.
Additional data are included with higher content of CNT+CMF, showing the trend and relationship is maintained, that the electrical resistivity of the SCC with CNT+CMF decay drastically at the percolation range and stabilise at higher concentrations. These high contents (3%) were not considered in PZR tests due to the loss of self-compactness of the concrete with such high content of CNT+CMF
- The existing or prospected issues should also be given in conclusion parts.
The conclusions have been rewritten
Reviewer 6 Report
The purpose of this study is to presents the thermal fatigue performance of SCC containing CNT and CMF due to high temperatures and self-diagnostic capability. It is interesting and organized very well. Some points should however be precised before an eventual publication:
Abstract need to be rewritten to report about the main and new findings obtained in this paper briefly. Please explain the similarities and differences between the results and other researchers, and authors should clarify the aim and the novelty of the study.Author Response
Reviewer 6
- Abstract to be rewritten to report about the main and new findings obtained in this paper briefly. Please explain the similarities and differences between the results and other researchers, and authors should clarify the aim and novelty of the study.
The abstract has been rewritten reflecting the main and new findings of the work with SCC in comparison to existing experience from literature.
Round 2
Reviewer 3 Report
I would like to see more in depth evaluation of the SCC behavior at Elastic, viscoelastic, viscoplastic levels. Please check the following papers format for more information.
- Effect of Evotherm-M1 on Properties of Asphaltic Materials Used at NAPMRC Testing Facility. Journal of Testing and Evaluation, 48(3).
- Characterization and validation of the nonlinear viscoelastic-viscoplastic with hardening-relaxation constitutive relationship for asphalt mixtures. Construction and Building Materials, 216, 648-660.
- A straightforward procedure to characterize nonlinear viscoelastic response of asphalt concrete at high temperatures. Transportation Research Record, 2672(28), 481-492.
Author Response
- I would like to see more in depth evaluation of the SCC behavior at Elastic, viscoelastic, viscoplastic levels. Please check the following papers format for more information.
Effect of Evotherm-M1 on Properties of Asphaltic Materials Used at NAPMRC Testing Facility. Journal of Testing and Evaluation, 48(3).
A straightforward procedure to characterize nonlinear viscoelastic response of asphalt concrete at high temperatures. Transportation Research Record, 2672(28), 481-492.
The authors certainly appreciate the comment and suggestion of the referee. We have read and analysed the references suggested by the reviewer. However due to the differences in the type of material, SCC and asphalt, and also the actual scope of the work carried out makes not feasible the implementation of the suggestion of the reviewer. Probably with another type of strategy and more specific study oriented to characterize nonlinear viscoelastic response of concrete at high temperatures could be possible to apply the model already proposed by for asphalt. From the experience of the authors on fire at high temperature this type of mechanical analysis for concrete at high temperature, even for fire events that concentrate most of the knowledge of mechanical performance of concrete, is not commonly addressed. More in detail:
Respect to Visco-(Plastic-elastic) response: Unfortunately in the SCC as in the Ordinary Potland Concrete this type of studies is not habitual, since the values of the elastic modules are very high unlike the asphaltic concretes, This type of study of the concrete is very complex and mainly it is applicable in the concrete for structures pre-stressed or they are related in systems with slow creep with sustained loads of long term duration, that it is not the case to evaluate in this work.
SCC elastic behavior: The SCC in the PZR tests is used below its elastic limit. As it is a non-destructive test, we are interested in having as a reference the Self-sensing property within this limit, but the elastic behaviour of the material is only included as a reference of the PZR test.
Reviewer 4 Report
At the beginning I would like to thank the authors for all the corrections. The manuscript is definitely clearer, especially the description of the research procedures is very accurate.
I am satisfied with most of the amendments, I would like to thank for the detailed Authors responses. There are two issues that still need to be developed. I understand that a detailed reference to them would require additional research, so I suggest that the authors at least pay attention to them in the conclusions.
The first issue is temperature. Figure 7 adds 105°C, which (as Authors wrote) “reflect the increase of the electrical resistivity due to the loss of liquid water”. Nothing is known about the impact of micro-cracks at the tested temperatures of 550°C. I propose to add in conclusions the information that concrete has not been tested at such temperatures, but there are such plans for the future + the part of Authors response to my comment “The consequences of temperature increase in PZR needs for a more sophisticated test system that warrantees the maintenance of T during test. The authors appreciate this comment to address this aspect in future.”
Second question is the repeatability of the measurement result in subsequent load cycles. I do not quite agree that it is a voltage issue. In figure 8 the increase in final FCR value compared to the beginning value is similar regardless of voltage. This problem should also be commented on in the conclusions as a deficiency of the described model.
Other proposed amendments:
Figure 7 - Please add to the legend the triangles related to 105°C,
Figure 8 - It is impossible to distinguish between types of lines - it is better to show them in colors,
Line 257-263 – change „Mpa” into „MPa”.
Author Response
- At the beginning I would like to thank the authors for all the corrections. The manuscript is definitely clearer, especially the description of the research procedures is very accurate.
I am satisfied with most of the amendments, I would like to thank for the detailed Authors responses. There are two issues that still need to be developed. I understand that a detailed reference to them would require additional research, so I suggest that the authors at least pay attention to them in the conclusions.
The first issue is temperature. Figure 7 adds 105°C, which (as Authors wrote) “reflect the increase of the electrical resistivity due to the loss of liquid water”. Nothing is known about the impact of micro-cracks at the tested temperatures of 550°C. I propose to add in conclusions the information that concrete has not been tested at such temperatures, but there are such plans for the future + the part of Authors response to my comment “The consequences of temperature increase in PZR needs for a more sophisticated test system that warrantees the maintenance of T during test. The authors appreciate this comment to address this aspect in future.”
The suggestion of the reviewer has been taken into account and introduced the comment both in the text (lines 449-451) and also highlighted in conclusions
- Second question is the repeatability of the measurement result in subsequent load cycles. I do not quite agree that it is a voltage issue. In figure 8 the increase in final FCR value compared to the beginning value is similar regardless of voltage. This problem should also be commented on in the conclusions as a deficiency of the described model.
Comments clarifying more the aspect of repeatability are included in lines 476-480 and highlighted in conclusions
- Figure 7 - Please add to the legend the triangles related to 105°C,
The figure has been modified in the new version
- Figure 8 - It is impossible to distinguish between types of lines - it is better to show them in colors,
A new color figure has been included
- Line 257-263 – change „Mpa” into „MPa”.
This has been corrected in the text
Author's Reply to the Review Report (Reviewer 5)
- The paper was properly revised.
The authors appreciate the reviewer found not new changes needed
Reviewer 5 Report
The paper was properly revised.
Author Response
- The paper was properly revised.
The authors appreciate the reviewer found not new changes needed